# Epidemiology and Risk Factors for Notifiable Scrub Typhus in Taiwan during the Period 2010–2019

**DOI:** 10.3390/healthcare9121619

**Published:** 2021-11-23

**Authors:** Fu-Huang Lin, Yu-Ching Chou, Wu-Chien Chien, Chi-Hsiang Chung, Chi-Jeng Hsieh, Chia-Peng Yu

**Affiliations:** 1School of Public Health, National Defense Medical Center, Taipei City 11490, Taiwan; noldling@ms10.hinet.net (F.-H.L.); trishow@mail.ndmctsgh.edu.tw (Y.-C.C.); chienwu@ndmctsgh.edu.tw (W.-C.C.); g694810042@gmail.com (C.-H.C.); 2Department of Medical Research, Tri-Service General Hospital, Taipei City 11490, Taiwan; 3Graduate Institute of Life Sciences, National Defense Medical Center, Taipei City 11490, Taiwan; 4Taiwanese Injury Prevention and Safety Promotion Association, Taipei City 11490, Taiwan; 5Department of Health Care Administration, Asia Eastern University of Science and Technology, New Taipei City 22061, Taiwan; fl004@mail.aeust.edu.tw

**Keywords:** epidemiology, climate, scrub typhus, *Orientia tsutsugamushi*, domestic, imported

## Abstract

Scrub typhus is a zoonotic disease caused by the bacterium *Orientia tsutsugamushi*. In this study, the epidemiological characteristics of scrub typhus in Taiwan, including gender, age, seasonal variation, climate factors, and epidemic trends from 2010 to 2019 were investigated. Information about scrub typhus in Taiwan was extracted from annual summary data made publicly available on the internet by the Taiwan Centers for Disease Control. From 2010 to 2019, there were 4352 confirmed domestic and 22 imported cases of scrub typhus. The incidence of scrub typhus ranged from 1.39 to 2.30 per 100,000 from 2010–2019, and peaked in 2013 and 2015–2016. Disease incidence varied between genders, age groups, season, and residence (all *p* < 0.001) from 2010 to 2019. Risk factors were being male (odds ratio (OR) =1.358), age 40 to 64 (OR = 1.25), summer (OR = 1.96) or fall (OR = 1.82), and being in the Penghu islands (OR = 1.74) or eastern Taiwan (OR = 1.92). The occurrence of the disease varied with gender, age, and place of residence comparing four seasons (all *p* < 0.001). Weather, average temperature (°C) and rainfall were significantly correlated with confirmed cases. The number of confirmed cases increased by 3.279 for every 1 °C (*p* = 0.005) temperature rise, and 0.051 for every 1 mm rise in rainfall (*p* = 0.005). In addition, the total number of scrub typhus cases in different geographical regions of Taiwan was significantly different according to gender, age and season (all *p* < 0.001). In particular, Matsu islands residents aged 20–39 years (OR = 2.617) and residents of the Taipei area (OR = 3.408), northern Taiwan (OR = 2.268) and eastern Taiwan (OR = 2.027) were affected during the winter. Males and females in the 50–59 age group were at high risk. The total number of imported cases was highest among men, aged 20–39, during the summer months, and in Taipei or central Taiwan. The long-term trend of local cases of scrub typhus was predicted using the polynomial regression model, which predicted the month of most cases in a high-risk season according to the seasonal index (1.19 in June by the summer seasonal index, and 1.26 in October by the fall seasonal index). The information in this study will be useful for policy-makers and clinical experts for direct prevention and control of chigger mites with *O. tsutsugamushi* that cause severe illness and are an economic burden to the Taiwan medical system. These data can inform future surveillance and research efforts in Taiwan.

## 1. Introduction

The *Rickettsia* spp. are widespread all over the world and pose an increasingly serious threat to human health due to frequent international traffic, the global greenhouse effect, climate change, and other factors [1]. Scrub typhus, caused by infection with *Orientia* species, is the most common rickettsial disease in the world. Scrub typhus, also called tsutsugamushi disease, is an acute infectious disease caused by *Orientia tsutsugamushi*, a Gram-negative bacterium that is intracellular parasite. It is transmitted to humans by the bite of certain kinds of trombiculid mites, or chiggers [2,3,4]. The chigger mite is also known as the red mite, so scrub typhus is also called red mite disease. Hatori first reported scrub typhus in Taiwan in 1908 [5]. The geographical distribution of scrub typhus is quite widespread across the globe. The area in which scrub typhus is found has been dubbed “**the Tsutsugamushi Triangle**”, and includes Japan to the northeast, northern Australia to the south, and Pakistan and Afghanistan to the northwest. Only *O. tsutsugamushi* causes scrub typhus in the Tsutsugamushi Triangle, and that other species of *Orientia* cause scrub typhus in other locations, such as South America, Africa, and the Middle East. The epidemic regions include South Korea, China, Japan, Taiwan, and the Philippines. Most cases occur in the rural areas of Southeast Asia, Indonesia, China, Japan, India, and northern Australia. It is spread across both the temperate and tropical zones of the world, and even the Himalayas at an altitude of more than 3000 m [6]. People are often infected in scrub typhus islands, where *O. tsutsugamushi* and rodents coexist. The chance of becoming infected with *O. tsutsugamushi* is associated with specific occupations and activities in infected areas. When susceptible persons (such as military personnel) enter an area with endemic scrub typhus, about 20–50% of them will become ill after several weeks or months [7]. In addition, (1) agricultural workers, (2) livestock workers, (3) forestry workers, (4) environmental services workers and (5) military personnel are high-risk groups for tsutsugamushi diseases because they work in fields, grasses or woods [8,9,10,11,12,13].

Scrub typhus is spread by the chigger stage of infected mites, which belong to the class Arachnida. The larvae are quite small and almost invisible to the naked eye. At present, about 3000 kinds of chiggers are recorded globally, but the chiggers that most often transmit human scrub typhus belong to the *Leptotrombidium* genus. In Taiwan, *Leptotrombidium deliense* is the main vector of scrub typhus. The animal hosts of *O. tsutsugamushi* infection include rodents and mammals (sheep, pigs, dogs and cats), as well as birds. The mites’ hosts are rodents, including rats [14]. Transmission occurs when an infected rat is bitten by a chigger mite, which in turn bites an uninfected rat or human. Chiggers infected with *O. tsutsugamushi* can pass *Orientia* through transovarial transmission, and carry *Orientia* in seven developmental stages: egg, deutovum (or prelarva), larva, nymphochrysalis, nymph, imagochrysalis and adult. The life history of a chigger is a single stage in a mite life cycle, which is greatly affected by temperature and humidity. The most suitable temperature for their development is 20–30 °C, and they favor higher humidity. The time required for a complete generation of *L. deliense* is about 59–135 days (average 89 days). The 3-month (89 day) generation time may correlate with peak summer and fall months. Chigger mite larvae are parasitic, so they are the vector of *O. tsutsugamushi* infection in humans [15]. The main natural hosts of chiggers are mice. Because mice often appear in grasses, woods and other places, chiggers also live in these areas and bite humans who touch the grass. The disease is not directly transmitted from person to person. According to studies in Taiwan, *O. tsutsugamushi* has a variety of rodent hosts. At present, *Rattus tanezumi*, *R. exclans*, *R. losea* and *Bandicota indica* are the main rodent hosts for the infected mites. *Rattus tanezumi* is easy to find in environmental areas where there are human activities (in houses, etc.) [16]. In addition, based on the survey of important rodent-borne infectious diseases in Taiwan, *R. tanezumi* had the highest prevalence of *O. tsutsugamushi* infections (100%), followed by *R. losea* (61.3%), *Bandicota indica* (45.8%) and *R. exclans* (34.6%). *R. tanezumi* had the highest seropositivity rate (91.7%), and *O**. tsutsugamushi* had a positivity rate of 95.8%; TP0607a and Karp were the most widely distributed OT strain and genotype (33.6%) [16]. The incubation period is about 1–3 weeks after being infected with *O**. tsutsugamushi* [17], and the symptoms that may appear are as follows:(1)High fever and shivering: after a week of fever, the body cadres may have a dark red papule, which expands to the limbs, and disappears after a few days.(2)Painless eschar is produced in bitten areas; sometimes it is not easy to find the eschar. An eschar, which usually begins as a primary papular lesion later crusts to form a black ulcer with central necrosis, is a distinct feature of scrub typhus. However, the possibility of *O. tsutsugamushi* infection cannot be excluded when the eschar is not found. According to a study conducted on patients in Taitung MacKay Memorial Hospital in Taiwan, only 69% of cases were found to have eschar [18].(3)Because of the short mouthparts of chiggers, they prefer to bite the soft parts of the skin such as the waist, genitals, underarms, breasts and around the eyes [19]. Therefore, to find eschars, these areas need to be inspected carefully.(4)Other symptoms include headache, night sweats, cough, muscle pain, conjunctival congestion and lymphadenopathy near the bite.(5)Severe cases may progress to pneumonia, acute myocarditis, acute liver and kidney failure or even death, with the mortality ranging from 1% to 70%. However, tsutsugamushi disease is endemic. People living in affected areas may be infected with tsutsugamushi disease many times, and depending on the strain of *O. tsutsugamushi* subsequent infections can be quite severe and even life threatening if not treated properly.

*Rickettsia* is not easy to diagnose clinically, as infection by many pathogens cause similar symptoms that are difficult to distinguish. The misdiagnosis or delay of the right treatment time of scrub typhus may cause a decline in the quality of life of patients, causing a further threat to their health and wasted medical resources.

Taiwan is located at 23°4′ north latitude and 121°0′ east longitude and has a subtropical climate. The average monthly temperature ranges from 16 °C to 29 °C, and the relative humidity ranges from 75% to 90%. Taiwan has become a developed country with a *per capita* gross domestic product (GDP) of US$27,437. Although there are effective treatment methods, cases related to scrub typhus infection still exist in Taiwan [20], which indicates that the effectiveness of epidemic prevention measures in controlling or eliminating the disease may be limited. Cases of scrub typhus occur in Taiwan throughout the year, most of which are sporadic. Clusters of cases are rare; the number of cases reported to be imported from overseas is also small. However, to date, little epidemiological information has been mined from big data to explore associations with the risk of scrub typhus in Taiwan. The aim of this study was to use the Taiwan National Infectious Disease Statistics System (TNIDSS) to investigate the epidemiological characteristics, including the number of domestic and imported cases, gender, age, season, differences in geographical location and trends of *tsutsugamushi* in the Taiwanese population from 2010 to 2019.

## 2. Materials and Methods

### 2.1. Ethical Policy

This study did not require ethical approval because it involved information freely available in the public domain. The analysis was performed on open-source datasets in which data are properly anonymized [21]. The authors are certain about the added value of this study, which conforms with the public use of the government reports.

### 2.2. Definition of Cluster Infections

Cluster infections refer to an aggregation of cases grouped in place and time that are suspected to be greater than the number expected, even though the expected number may not be known [22].

### 2.3. Definition of Reported and Confirmed Cases

Notification is defined as a suspected patient who meets clinical conditions (clinical conditions are sudden with persistent high fever, headache, back pain, chills, night sweats, lymphadenopathy, painless eschar at the chigger bite and red skin with macular papules after 1 week, sometimes accompanied by pneumonia or abnormal liver function). In addition, those who test positive in any one of the following tests are defined as positive: (1) Clinical specimens (blood or skin wounds (eschar)) test positive for *O. tsutsugamushi* by nucleic acid detection; (2) Indirect immunofluorescene assay detects acute phase (or initial collection) serum, with a neutralization antibody titer of IgM of more than 1:80; IgM antibody has a potency of more than 1:80, and the IgG titer was more than 1:320; (3) Using the indirect immunofluorescence staining to detect matched (acute and convalescent) serum, a ≥4-fold increase in the IgG titer against *O.*
*tsutsugamushi* is observed. A confirmed case is defined as attaining a positive result [23].

### 2.4. Data Source

Taiwan has a population density of 627/km^2^, an area of 36,188 square kilometers, and a population of approximately 23.5 million. The majority (95%) of the population lives in western Taiwan. Data in this study are divided into the Taipei area and the Gaoping area (the southern region of Taiwan between Kaohsiung and Pingtung counties), along with northern, central and southern Taiwan. Only 5% of the population lives in eastern Taiwan, and is a vulnerable population with respect to medical care and socio-economic status. In addition, the three outlying islands of Penghu, Kinmen and Matsu were considered separate areas.

This study used the Taiwan National Infectious Disease Statistics System (TNIDSS), a publicly-available internet database established by the Taiwan Centers for Disease Control (Taiwan CDC) [20]. The database includes all statutory infectious diseases in categories 1 to 5 as stipulated by the Infectious Disease Prevention and Control Law. In order to ensure information security and prevent the leakage of case privacy, the system’s database does not store any case details, only secondary data with statistical values. Taiwan CDC has maintained an open infectious disease statistical data query system since 1996 to provide information on the number of confirmed cases of scrub typhus. It provides the public, academia and the media with the latest epidemic information about domestic infectious diseases including scrub typhus. Scrub typhus has been listed as a Category IV Notifiable Infectious Disease since 2007 in Taiwan based on the Communicable Disease Control Act. The open data include the number of confirmed cases of scrub typhus, the date of receipt, the date of onset, the date of diagnosis by the Department of Health, and the number of local or cases imported from overseas from January 2010 to December 2019. The internet databases do not contain the medical history of patients, signs and symptoms, or results of laboratory tests. In addition, this study also uses the publicly-available internet database maintained by Taiwan’s Central Weather Bureau [24]. The open data include temperature, rainfall, relative humidity, hours of sunshine and station pressure from January 2010 to December 2019.

### 2.5. Data Analysis

This study confirmed the number of local and imported cases of tsutsugamushi disease between 2010 and 2019, and investigated the differences in their epidemiological characteristics (gender, age, date of disease, region of residence). Descriptive data are shown as means and summary statistics where appropriate. Categorical variables were compared using the chi-square test. Multiple linear regression analysis was used to test the relationships between atmospheric pressure, temperature, relative humidity, rainfall and hours of sunshine, and the number of confirmed cases. In addition, we also used the polynomial regression method to predict the long-term trend of the number of local cases of scrub typhus in Taiwan, and the trend or seasonal effects, to estimate the ratio by calculating a moving average and obtaining the seasonal index. All statistical analyses were performed using SPSS software (IBM SPSS Statistics 21; Asia Analytics Taiwan, Taipei, Taiwan). All statistical tests were two-sided with an α level of 0.05. *p* values of less than 0.05 were considered statistically significant.

## 3. Results

### 3.1. Study Population

During the study period (January 2010–December 2019), 4374 confirmed cases of scrub typhus were recorded (Table 1). There were 4352 domestic cases and 22 imported cases.

### 3.2. Epidemiological Features

(1) Table 2 shows the difference in gender, age, season, and region of residence among the confirmed cases for each year. The confirmed cases of scrub typhus numbered 2699 males and 1675 females. There was a significant difference in gender from 2010 to 2019 (*p* < 0.001). There were 332 cases ≤ 19 years old, 1183 cases 20–39 years old, 2156 cases 40–64 years old and 703 cases ≥ 65 years old. The distribution of cases in age groups were not equal (*p* < 0.001). There were 659, 1684, 1262 and 769 cases in spring, summer, autumn and winter, respectively. The case number in each season showed a discrepancy (*p* < 0.001). The confirmed cases of scrub typhus numbered 484 in the Taipei area, 252 in northern, 413 in central and 128 in southern Taiwan, as well as 556 in the Gaoping area, 1191 in eastern Taiwan, 683 in the Penghu islands, 507 in the Kinmen islands and 160 in the Matsu islands. There was a disparity in their place of residence (*p* < 0.001).

(2) Table 3 shows the differences in gender, age, and region of residence among the confirmed cases with seasonal variation. Confirmed cases of scrub typhus in males numbered 379 in spring, 1016 in summer, 767 in autumn and 537 in winter. There was an association between season and gender (*p* < 0.001). There were 309, 781, 644 and 422 cases among males in spring, summer, autumn and winter, respectively, in the 40–46 age group. The case numbers across seasons showed a discrepancy with age group (*p* < 0.001). In eastern Taiwan, confirmed cases of scrub typhus numbered 238 in spring, 324 in summer, 324 in autumn and 324 in winter. There was a disparity between season and place of residence (*p* < 0.001).

(3) Table 4 shows a comparison of gender, age and season among the confirmed cases in the different regions. Confirmed cases of scrub typhus among males numbered 300 in Taipei, 145 in northern, 255 in central and 84 in southern Taiwan, 356 in the Gaoping area, 757 in eastern Taiwan, 365 in the Penghu islands, 325 in the Kinmen islands and 112 in the Matsu islands. There was an association between the area and gender (*p* < 0.001). There were 216, 141, 207, 62, 317, 635, 345, 182 and 51 males in the Taipei area, northern area, central area, southern area, Gaoping area, eastern area, Penghu islands, Kinmen islands and Matsu islands in the 40–46 age group, respectively. The case number in regions showed a discrepancy between age groups (*p* < 0.001). The confirmed cases of scrub typhus included 148 in the Taipei area, 76 in the northern area, 146 in the central area, 55 in southern area, 238 in the Gaoping area, 324 in the eastern area, 245 in the Penghu islands, 248 in the Kinmen islands and 105 in the Matsu Islands in the summer. There was a disparity between place of residence and season (*p* < 0.001).

(4) Trend in the distribution of confirmed cases.

Incidence rate per 100,000 people peaked in 2013 and 2015–2016 during the study period (Figure 1A). Males 60–69 years old had the highest incidence rate according to gender and age group (Figure 1B,C). The 50–59 age group had a high incidence rate in both males and females (Figure 2). Among the regions of residence, the incidence rate of scrub typhus was highest in the Matsu Islands (Figure 3A) every year. The main outbreak areas of confirmed scrub typhus cases were eastern Taiwan, the Penghu islands and the Kinmen islands (Figure 3B).

(5) Distribution of the number of imported cases according to gender, age, season and region of residence. Most imported cases were 16 males (72.7%, 16/22). Of the imported cases, the highest numbers in each category were: nine aged 20–39 (40.9%, 9/22), eight in summer (36.4%, 8/22) and seven living in Taipei (31.8%, 7/22) (Table 1).

(6) With regard to other climatic factors, temperature was significantly correlated with the number of confirmed cases, β = 0.626, *p* = 0.005; rainfall was also significantly correlated with the number of confirmed cases, β = 0.298, *p* = 0.010. R^2^ = 0.417, Adj R^2^ = 0.391, F value = 16.296, df = (5, 114), as shown in Table 5.

### 3.3. Long-Term Trend Prediction and Seasonal Index

(1) The long-term trend of local cases of tsutsugamushi disease is shown in Figure 4, the polynomial regression equation is y = −0.0927x^5^ + 3.159x^4^ − 38.476x^3^ + 199.15x^2^ – 393.85x + 619.47

(2) The seasonal index of local cases of scrub typhus is shown in Table 6. The seasonal index for summer was 1.193 in June, 1.187 in July and 0.62 in August. The seasonal index for autumn was 1.012 in September, 1.263 in October and 0.725 in November.

## 4. Discussion

Scrub typhus is a highly infectious disease of *O. tsutsugamushi*. Humans can be infected through being bitten by infected chiggers. According to the statistics used for this study, the total number of confirmed cases of scrub typhus was 4374 in Taiwan from 2010 to 2019. Many factors affected the number of cases of scrub typhus. Among them, a suitable environment, the coexistence of chiggers, *O. tsutsugamushi* and rodents are concluded to be the four indispensable elements for scrub typhus incidence. Among them, the environment has considerable influence on the number and distribution of the chiggers and rodents [25]. The activity of chiggers is related to environmental factors, such as temperature and humidity. Temperature was the main factor in each season [26]. Our study showed that the number of cases in summer and autumn was significantly different compared with other seasons, which is similar to the results of studies conducted in other countries [27]. It is inferred that the rise of temperature in a season may be one of the reasons for the increase in scrub typhus cases. In addition, in this study, the multiple regression method was used to analyze the number of confirmed cases, temperature and rainfall. We found that higher temperatures or greater volumes of rainfall were associated with a greater number of confirmed cases. Temperature and rainfall are risk factors for scrub typhus. The highest incidence of tsutsugamushi disease was 2.30 per 100,000 population in 2013.

There were incidence peaks in 2013 and 2015–2016. The possible reason for this is that the temperature in the autumns of 2013 and 2015–2016 was significantly higher than that in other years, which may have increased the number of cases of scrub typhus. The number of confirmed cases in the local area began to rise rapidly from April to May. Most of the cases were found in summer and autumn. One peak was reached from June to July in the summer, and another peak from September to October in the autumn. The reason may be that the temperature rose to about 29 °C and the summer climate is too hot and slows down the activity of chiggers. However, the temperature drops in autumn and is suitable for chigger growth [28,29]. Over the last 5 years, the peak for total cases was in 2015–2016. This made it difficult to control the outbreak and cost considerable medical resources. In 2016, the climate was extreme; the hottest year in the world in over one hundred years was recorded, which is reflected in this study. The global average temperature in this year was 0.94 °C higher than the average temperature in the 20th century. Rainfall was higher in spring and the temperature was extremely high in summer and autumn in Taiwan in 2016. The weather was warm with less rain in December and the annual average temperature was 24.4 °C, which was the second highest annual average temperature since 1947 [30,31]. The findings of this study suggest that climate change is an important factor affecting infectious diseases, and it is worthy of attention from public health and epidemic prevention experts, with early planning of response measures suggested to reduce the risk of disease.

The results of our study showed that male cases account for a larger proportion of scrub typhus cases. The number of male cases was about 1.5–2 times that of female cases, which is opposite to studies conducted in other countries [32]. In this study, the distribution of the confirmed cases in each age group was explored by gender. This showed that age (especially 40–69 years old) is indeed one of the disease risk factors in gender. Increase risk seen in 20–29 year-old men was an exception. The increase in this age group in Taiwan may be associated with occupation (such as military personnel) and outdoor activity. Most cases were over the age of 20 at onset, and the 40–69 age group accounted for more than 50% of cases. These results were similar to the other studies [33]. This study indicated that the male population of 40–69 years old who engaged in outdoor or mountainous activities was associated with a high incidence rate of scrub typhus. The distribution of gender differences was also explored. Although the results showed that for both males and females the peak of the total number of cases was between the ages of 50–59, the highest incidence was at the age of 60–69. Self-health management of these populations and media education is suggested as a government epidemic prevention policy to slow down the spread of the disease.

The high risk areas for scrub typhus were found to be almost all in the eastern part of Taiwan, except from 2013 to 2016, when the highest proportion of cases were in the Penghu islands. According to the previous Taiwanese studies, there is not only a higher density of chigger mites, but also higher antibody-positive rate (70%) of *O. tsutsugamushi* in the serum of the small rodent hosts in these areas. This finding highlights that the rodent hosts carry the maximum number of chiggers between October and November in eastern Taiwan [34]. Therefore, the eastern region has a high incidence area of scrub typhus, and may be a risk area for *tsutsugamushi* bites all the year round. The geographical area of Hualien and Taitung counties is large, and agricultural land makes up as much as 75% of the whole region. Most of the cases are in different risk areas. The findings of this study suggest that the risk of scrub typhus increased when people were engaged in agriculture or outdoor activities. The previous study showed that the annual number of scrub typhus cases was correlated with the percentage of rats carrying chigger infestation in the Penghu islands from 2003 to 2015 (r = 0.74, *p* = 0.022). A previous study succeeded in predicting the number of cases in the Penghu islands and could be used as a reference for epidemic prevention [35]. It is worth noting that the Kinmen islands are a tourist attraction and an area that Chinese tourists must pass through. The island was a risk area for scrub typhus before 2014, but the cases of scrub typhus decreased significantly in the three years from 2017 to 2019. The government’s tourism policy has reduced the number of Chinese tourists since 2016, resulting in a large decline in the economy and tourism activities in Kinmen islands; as a result, the local people may have reduced contact with vectors. This may suggest that the implementation of government policies and barrier measures will still affect the effectiveness of epidemic prevention. In summary, our analysis confirmed that eastern Taiwan and the Penghu Islands were still high risk areas for scrub typhus.

From the above discussion, it can be clearly seen that people living in high-risk areas are indeed key factors in scrub typhus infection. Therefore, the relationship between susceptible populations, gender, age, season and residence was explored. The most common occurrence of scrub typhus was found to be among males between 40–64 years old, and the epidemic prevailed in summer, but there are two more subtle findings worthy of particular note. First, the Matsu islands accounted for the highest proportion of cases aged 20–29 (40%, 64/160). The Matsu islands constitute a front line for the military, and soldiers are mostly young men. They often work in grass so the risk of the disease increases sharply. Secondly, the winter incidence remained high in the Taipei area, northern Taiwan and eastern Taiwan. This result might be attributed to: (1) the climate being warm in winter because of global warming, and (2) infectious vectors and wild rodents being frequently active. The winter weather was often rainy in Taipei and northern Taiwan, so people often take trips to the warmer eastern region which has many national tourist attractions. The eastern area was originally a high-risk area for scrub typhus, so that the epidemic situation extended from summer and autumn to winter, which became the hidden worry of public health disease prevention and control.

If scrub typhus is untreated, the mortality rate can be as high as 70%, and if properly treated, the mortality rate can be reduced to less than 5% [36]. According to the literature [37], deaths from scrub typhus have been rare in Taiwan since 2001. A total of 4529 persons were diagnosed from 2001 to 2012, and only four cases were directly related to a cause of death. There were two and one deaths in 2013 and 2016, respectively. One case was a 37-year-old French female. She entered Taiwan in mid-April 2016, and developed fever (≥38° C) and chills at the end of April after traveling to Orchid Island for a week [37]. Her sickness deteriorated rapidly. Emergency treatment was ineffective, and the patient died after medical diagnosis and hospitalization. The reasons that the physicians failed to use antibiotic treatment in time may be because they did not ask about TOC (travel history, occupation, contact) or think about scrub typhus as a possible diagnosis. The number of foreign tourists visiting Taiwan has increased in recent years. The threat of the import of scrub typhus from neighboring areas, such as China and Southeast Asia, which are endemic areas for scrub typhus, still exists [38]. Twenty-two cases have been imported since 2010. Most of the cases were males, 20–39 years old, infected in summer, and living in Taipei or the central region. Therefore, in order to avoid or prevent the spread of imported scrub typhus and its effect on the health of local people, it is recommended that the disease is prevented from entering from overseas, by setting up strict epidemic prevention measures at airports and docks. In addition, clinicians are suggested to inquire about the travel histories of foreign tourists and patients with a domestic travel history that have fever or suspected infections, by obtaining complete Travel, Occupation, Contact, and Cluster information (TOCC). They should also prioritize the identification of cases that are related to rickettsial diseases, such as scrub typhus, to reduce mortality and stop the spread of the disease by early notification and early treatment.

The cases of scrub typhus in Taiwan are generally sporadic, and clustering events are rare. In January 2017, people who traveled to Sandimen Township in Pingtung County led to a cluster of scrub typhus [39]. The index case was a 41-year-old male who developed fever and skin eschar symptoms in mid-January 2017. He was admitted to hospital in Pingtung County at the end of January, and notified of the contraction of scrub typhus in early February. He was confirmed to have scrub typhus by IgM, IgG and PCR detection. After the indicator case was diagnosed, the health department immediately investigated the suspected infection region. They revealed that a total of 18 people had accompanied the man in a visit to Ta-Mu-Mu-Shan Mountain in Sandimen Township in mid-January. Six of them had suspected symptoms and were subsequently diagnosed with scrub typhus. The literature deduces that the co-infection occurred at Ta-Mu-Mu-Shan Mountain in Sandimen Township. This cluster event represents a rare mountain tourism cluster outbreak in Taiwan, and it occurred earlier in the year than scrub typhus outbreaks in other years. Based on the data provided by the Taiwan Weather Bureau, it is speculated that the cluster event was affected by climate variation. There was a warm winter at the end of 2016 to the beginning of 2017, and this phenomenon possibly increased the activity of infectious *O. tsutsugamushi* and caused the spread of the disease.

Zoonoses are any infectious diseases that can be transmitted from animals to humans or from humans to animals, directly spread between humans and animals, or can spread by vectors, bringing pathogens (such as viruses or bacteria) into another organism. The route of transmission is through contact, inhalation or ingestion of food and water containing pathogens or bites with pathogenic vectors, which endanger human health [40,41]. *O. scrub typhus**is* is maintained in nature through both human urban and wildlife environmental cycles involving chigger mite vectors and human or vertebrate animal hosts. In this study, the epidemiological meaning of scrub typhus transmission of zoonotic diseases was investigated.

Population growth, and excessive reclamation and use of land have destroyed natural spaces and disturbed the ecosystem in Taiwan so that human contact with wild animals and disease vectors has increased. Therefore, residents in Taiwan’s metropolitan or rural areas (such as the eastern region) and wild animals in the mountainous or suburban areas usually live together in the same natural geographic area. In fact, the natural geographic area is a new hotspot of modern urban living space, which is caused by arthropod vectors. According to the literature, *O. tsutsuganushi* or *Borrelia burgdorferi* mediated by arachnids are often disseminated between humans and mice, which accelerate the outbreak of these cross-species-spreading diseases and put humans at risk of pandemics and serious threats to health [42,43]. It is recommended that: (1) In the face of diseases caused by *O. tsutsuganushi*, in addition to understanding the way the disease is transmitted, a comprehensive approach to disease prevention and control should be adopted. Strengthening environmental protection and preventing vector breeding may effectively stop outbreaks and protect citizens from public health crises; (2) To prevent the recurrence of a pandemic, governments or medical and public health expert groups around the world should continue to maintain vigilant disease surveillance, scientific research and education of the general population. This is the best weapon to fight disease outbreaks, and it is also the best tool for mitigating epidemics, the only way to eliminate infectious diseases common to humans and animals.

In Taiwan, people who visit the tombs of their ancestors to clean the grave sites, go out to the suburbs, camp or engage in mountaineering activities, etc., are more likely to come into contact with grassy environments where chiggers breed, increasing the risk of contracting scrub typhus. There is no effective vaccine for scrub typhus at present, and taking drugs for pre-prevention is not recommended [44]. There are three types of immunity: vaccine immunity, natural immunity and behavioral immunity. It is estimated that “behavioral immunity” will be the main effective epidemic prevention tool in the case of scrub typhus. Therefore, it is recommended that the public adopt the following preventive health care measures to reduce the risk of illness: (1) When traveling or working in the countryside, entry into grassy areas should be limited and “self-protection” measures implemented. If entering grassy areas is needed, protective clothing such as long-sleeved pants, boots and gloves should be worn to avoid skin exposure; (2) Mosquito repellent (including diethyltoluamide [45]) approved by the Ministry of Health and Welfare should be applied to exposed parts of the body, and used according to the instructions to avoid being bitten by chigger mites; (3) After leaving the grass, people should wash as soon as possible and change all clothes to reduce the chance of infection; (4) Weeds should be removed to eliminate chigger breeding grounds, and rodent holes and gaps near residences, on both sides of roads and bridges and other grassy areas should be sealed, and the home environment should be kept clean to avoid rodent breeding; (5) If symptoms of scrub typhus are seen, medical attention should be sought as soon as possible and physicians should be informed of travel history, or whether there is a history of exposure to suburbs, grass, etc., to provide a reference for the physician’s clinical diagnosis. Epidemic prevention work involves large-scale rodent hunting operations, which are an emergency prevention and control measure to prevent the spread of an epidemic. In addition, home or community environmental management can also be used to reduce the density of rats and chiggers, decrease the chance of exposure to human with rats or mites and strengthen public health education so that the public can understand the importance of preventing rodents or disease vectors and improve recognition of the disease and independent management of the environment. The density of rodents and chiggers can be cut off from the original source in this way. Scrub typhus is an infectious disease with a high incidence in Taiwan and once it breaks out, it often poses a threat to public health and has a great impact on society. Preventing scrub typhus as soon as possible is the key to controlling outbreaks quickly and effectively.

According to the literature, Chatterjee et al. used the polynomial regression method to establish a model and predict the incidence of malaria with a high confidence level in Chennai city, India [46]. In this study, we used the number of local cases of scrub typhus in Taiwan over the past 10 years to predict and analyze the long-term trend of future cases. Linear regression analysis is one of the most widely used bio-statistical techniques for long-term trend research. The validity of the regression model and its explanatory power are generally based on the coefficient of determination (R^2^), as an indicator. In regression analysis, the main function of R^2^ is to evaluate the explanatory degree of the change in the corresponding variables in the regression model; that is, R^2^ is an index to evaluate the quality of the regression model. R^2^ ≥ 0.75 indicates a high degree of interpretability; 0.75 > R^2^ ≥ 0.5 indicates a medium degree of interpretability; less than 0.5 indicates a low degree of interpretability [47]. According to the polynomial regression model of this study, its R^2^ is 0.51, which means that the long-term trend prediction model can be explained to a moderate degree. The possible reason is that climate variation and other factors have resulted in an increase in the annual local cases of scrub typhus in Taiwan over the last decade. Thus, the regression model could not reach a higher level (R^2^ > 0.75) to explain the changes in the total number of local cases each year, and this affected the function of prediction. It is recommended that monitoring of the impact of climate influence factors on the number of confirmed cases of scrub typhus is strengthened, to stabilize the long-term trend prediction and to establish the regression model with high R^2^ as an effective tool for predicting the number of scrub typhus cases in the future.

In addition, based on the periodic seasonal variations of scrub typhus that occur repeatedly every year, the monthly changes in the number of local cases of scrub typhus in Taiwan during the summer and autumn high-risk seasons was also explored. 

The higher the seasonal index, the higher the proportion of cases in that month of the season. In this study, the seasonal index was highest in June during summer, and October during autumn. This showed that the number of cases may be higher in June and October during high-risk seasons (summer and autumn) of scrub typhus. It is recommended that the government strengthen public health preventive measures in the above-mentioned months. People should also take measures when traveling to high-risk areas to strengthen physical protection or reduce the frequency of going out to avoid the spread of outbreaks caused by infection.

The “One World, One Health” concept has been advocated by the World Organization for Animal Health in recent years [48]. This means that the spread of disease has crossed national boundaries and can extend all over the world in a short period of time by air transportation. It is no longer sufficient for one country conduct self-surveillance and self-prevention against infectious diseases, but all countries are required to work together to control disease. In the natural environment, “One Health” refers to the close interdependence among humans, other animals and environmental health. From the perspective of scrub typhus (a zoonotic infectious disease), the larvae of trombiculid mites infected with *O. tsutsugamushi* bite rodents, causing disease in wild animals or chiggers with pathogens, and they may also be transmitted to humans. In addition, different human activities have caused environmental alteration and deterioration, causing a decline or lack of health in other organisms. Nature may counterattack and affect human society. In other words, every individual on the earth has its own role; everyone is intertwined. If a connection is broken, the operation of the ecosystem will be changed, and of course the survival of human beings will be affected. Forward-looking concepts of public health must become the basis for governments or public health experts to formulate health policies and decide the allocation of health resources.

This study has two drawbacks. First, the statistics about infectious diseases disclosed by the Taiwan CDC on the internet platform only provide basic epidemiological data about scrub typhus patients, with no clinical data. Therefore, this study cannot compare differences or trends in patient clinical symptoms. Second, the information disclosed on the platform also does not contain information about the genotype of *O. tsutsugamushi,* neither the kind of genotype of *O. tsutsugamushi* that is prevalent in Taiwan nor the genetic relationship when comparing the genotype of *O. tsutsugamushi* in this country with other countries remains unknown. However, this research has a unique advantage—that is, the information of the existing online public platform of the public sector in Taiwan is correct in real time. In addition, the database has stored data for many years, allowing researchers or institutions access to rich statistics and infectious disease data.

## 5. Conclusions

In conclusion, this study reveals the characteristics and trends of local and imported cases of scrub typhus from 2010 to 2019. The incidence of scrub typhus was 2.30 per 100,000 population in 2013, which was the highest during the study period. Cases among those aged 40–64 years exhibited a gradual increase in incidence and a distinct pattern of seasonal variation (summer and fall) during the study period. Furthermore, more male cases of scrub typhus were seen than female cases and the risk areas might be the Penghu islands and eastern area of Taiwan. This study also highlighted the increased risk to 20–29 year olds living in the Matsu islands, as well as temperature, rainfall and winter in the Taipei area, along with northern and eastern Taiwan. The polynomial regression model was used to successfully predict the long-term trend of local cases of tsutsugamushi disease in Taiwan, and the seasonal index was used to predict the number of high-risk months during the high-risk season. The seasonal index was highest in June during summer, and October during autumn. This information will be useful for policy-makers and clinical experts to direct prevention and control activities to chigger mites with *O.*
*tsutsugamushi* that cause severe illness and are a burden to Taiwanese. The data will inform future surveillance and research efforts in Taiwan.

## Figures and Tables

**Figure 1 healthcare-09-01619-f001:**
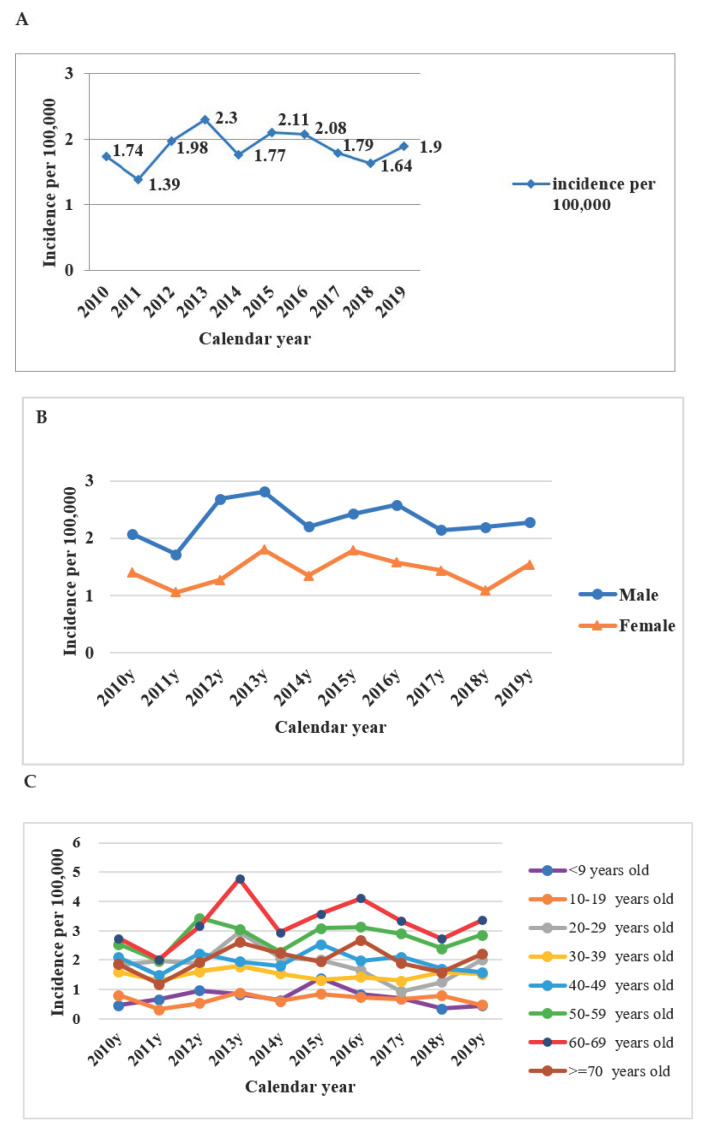
Incidence of confirmed scrub typhus among patients in Taiwan according to (**A**) population, (**B**) sex and (**C**) age by year from 2010 to 2019.

**Figure 2 healthcare-09-01619-f002:**
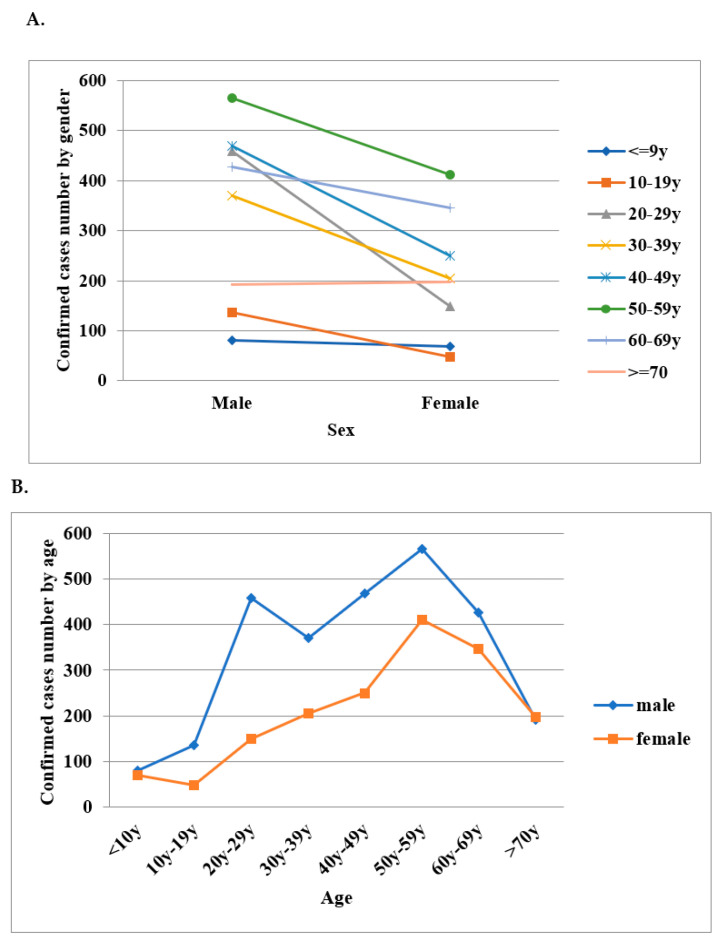
(**A**). Sex distribution dominated by age groups for domestic and imported patients with scrub typhus between 2010 and 2019. (**B**). Age distribution dominated by men and women for domestic and imported cases of scrub typhus between 2010 and 2019.

**Figure 3 healthcare-09-01619-f003:**
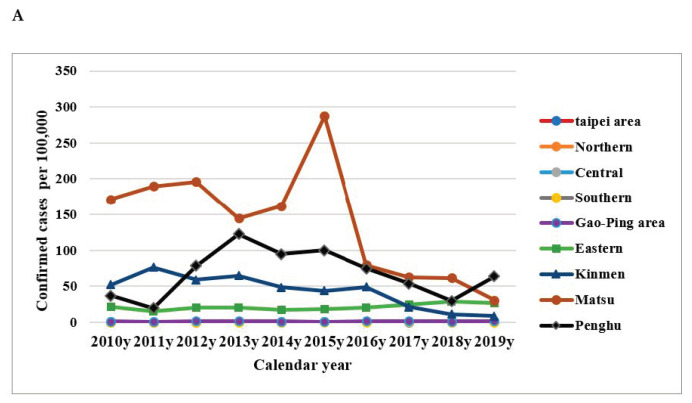
Major outbreak area of scrub typhus in Taiwan between 2010 and 2019. (**A**). Incidence of confirmed cases of scrub typhus in Taiwan according to region. (**B**). the major outbreak area is represented by the red line.

**Figure 4 healthcare-09-01619-f004:**
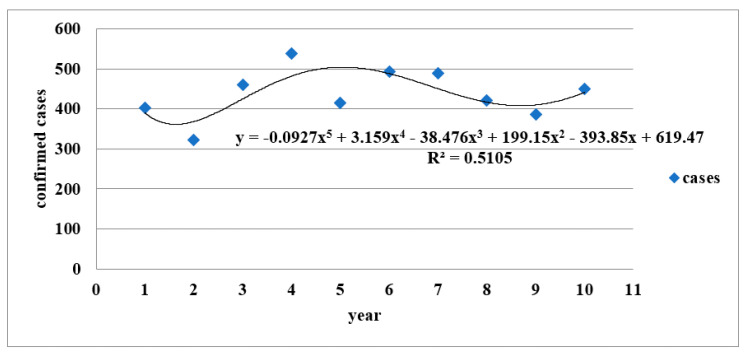
Polynomial regression prediction of domestic and imported cases with scrub typhus in Taiwan between 2010 and 2019. Numbers 1–10 on the *x*–axis represent 2010–2019.

**Table 1 healthcare-09-01619-t001:** Epidemiological features of domestic and imported cases of scrub typhus in Taiwan during 2010 and 2019.

Variables	All Cases	Domestic Cases	Imported Cases
	N = 4374	N = 4352	N = 22
Sex			
Male	2699	2683	16
Female	1675	1669	6
Age group			
<20	332	330	2
20–39	1183	1174	9
49–59	2156	2151	5
≥60	703	697	6
Year group			
2010–2014	2136	2129	7
2015–2019	2238	2223	15
Season			
Spring	659	656	3
Summer	1684	1676	8
Fall	1262	1255	7
Winter	769	765	4
Residency			
Taipei area	484	477	7
Northern	252	249	3
Central	413	407	6
Southern	128	127	1
Gaoping area	556	553	3
Eastern	1191	1189	2
Kinmen	507	507	0
Matsu	160	160	0
Penghu	683	683	0

**Table 2 healthcare-09-01619-t002:** The analysis of epidemiological features from a survey of domestic and imported cases of scrub typhus between 2010 and 2019 in Taiwan.

Variables	Domestic and Imported Cases										*p* Value
	2010/1–2010/12	2011/1–2011/12	2012/1–2012/12	2013/1–2013/12	2014/1–2014/12	2015/1–2015/12	2016/1–2016/12	2017/1–2017/12	2018/1–2018/12	2019/1–2019/12	
Imported cases											
Yes	1	1	2	1	2	2	4	0	2	7	*-*
No	401	321	458	537	412	492	484	421	384	442	
Sex											
Male	241	200	313	328	257	284	302	251	257	266	<0.001
Female	161	122	147	210	157	210	186	170	129	183	
Age											
<19	35	24	36	43	30	51	36	31	26	20	<0.001
20–39	125	117	126	167	128	116	108	79	99	118	
40–64	196	143	242	238	188	258	247	229	202	213	
≥65	46	38	56	90	68	69	97	82	59	98	
Season											
Spring	56	28	70	61	68	58	104	63	51	100	<0.001
Summer	123	174	191	238	186	157	147	170	130	168	
Fall	124	61	128	166	104	200	164	101	97	117	
Winter	99	59	71	73	56	79	73	87	108	64	
Residency											
Taipei area	65	33	40	47	27	72	58	38	35	69	<0.001
Northern	16	13	27	41	26	25	31	26	23	24	
Central	36	36	43	46	34	46	42	31	56	43	
Southern	18	10	6	13	5	17	14	21	5	19	
Gaoping area	41	29	68	60	49	38	74	76	56	65	
Eastern	123	86	114	116	97	101	116	136	156	146	
Kinmen	50	77	64	76	60	57	66	29	16	12	
Matsu	17	19	21	17	20	36	10	8	8	4	
Penghu	36	19	77	122	96	102	77	56	31	67	

**Table 3 healthcare-09-01619-t003:** Association between season and gender, age and region of residence from a survey of domestic and imported cases of scrub typhus between 2010 and 2019 in Taiwan.

Variables	Domestic and Imported Cases				*p* Value
	Spring	Summer	Fall	Winter	
Sex					
Male	379	1016	767	537	<0.001
Female	280	669	494	232	
Age					
<19	59	124	96	53	<0.001
20–39	177	496	306	204	
40–64	309	781	644	422	
≥65	114	284	215	90	
Residency					
Taipei area	70	148	82	184	<0.001
Northern	23	76	75	78	
Central	55	146	125	87	
Northern	17	55	34	22	
Gaoping area	81	238	170	67	
Eastern	238	324	324	305	
Kinmen	30	348	125	4	
Matsu	0	105	47	8	
Penghu	145	245	279	14	

**Table 4 healthcare-09-01619-t004:** Association between region of residence and gender, age and season from a survey of domestic and imported cases of scrub typhus between 2010 and 2019 in Taiwan.

Variables	Domestic and Imported Cases									*p* Value
	Taipei Area	Northern	Central	Southern	Gao-Ping Area	Eastern	Kinmen	Matsu	Penghu	
Sex										
Male	300	145	255	84	356	757	325	112	365	<0.001
Female	184	107	158	44	200	434	182	48	318	
Age										
<19	26	10	27	6	24	104	31	13	91	<0.001
20–39	180	87	135	45	146	237	158	77	118	
40–64	216	141	207	62	317	635	182	51	345	
≥65	62	14	44	15	69	215	136	19	129	
Season										
Spring	70	23	55	17	81	238	30	0	145	<0.001
Summer	148	76	146	55	238	324	348	105	245	
Fall	82	75	125	34	170	324	125	47	279	
Winter	184	78	87	22	67	305	4	8	14	

**Table 5 healthcare-09-01619-t005:** Association between confirmed cases and climate factors from a survey of patients with scrub typhus and climate factors between 2010 and 2019 in Taiwan.

Parameters	Non-Standardization Coefficient		Standardization Coefficient	*p* Value
	B Value	Standard Error	β Value	
Air temperature (°C)	3.279	1.146	0.626	0.005
Rainfall (mm)	0.051	0.019	0.298	0.010
Sunshine duration (h)	0.100	0.084	0.209	0.237
Relation humidity (%)	0.143	0.877	0.019	0.870
Station atmospheric pressure (hPa)	1.697	0.888	0.372	0.059
*R* ^2^	0.417			
*Adj R* ^2^	0.391			
*F*	16.296			
*df*	(5, 114)			

**Table 6 healthcare-09-01619-t006:** Seasonal index of summer and fall from a survey of domestic and imported cases of scrub typhus between 2010 and 2019 in Taiwan.

Month		Cases/Mean										Non-Adjusted Seasonal Index	Adjusted Seasonal Index
		2010 ^a^	2011	2012	2013	2014	2015	2016	2017	2018	2019		
Summer	Jun ^b^	0.000	1.247	1.386	1.144	1.168	1.490	1.481	1.266	0.773	1.272	1.123	1.193
	Jul.	1.537	1.086	0.927	1.155	1.177	0.936	0.939	1.182	1.315	0.911	1.117	1.187
	Aug.	0.526	0.701	0.676	0.796	0.439	0.644	0.496	0.826	0.732	0.000	0.584	0.62
Fall	Sep.	0.000	0.687	1.513	1.067	1.368	0.968	0.839	0.870	0.879	1.308	0.950	1.012
	Oct	1.452	1.328	1.008	1.436	0.981	1.305	1.061	1.010	1.268	1.000	1.185	1.263
	Nov	0.957	0.446	0.614	0.528	0.529	0.959	1.142	0.946	0.681	0.000	0.680	0.725

^a^: the rows represent “year”. ^b^: the columns represent June, July, August, September, October, November.

## Data Availability

Taiwan Centers for Disease Control. Taiwan National Infectious Disease Statistics System. Available online: https://nidss.cdc.gov.tw/ch/ (accessed on 1 July 2021).

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
