# Peer review of "Epidemiology and Risk Factors for Notifiable Scrub Typhus in Taiwan during the Period 2010–2019"

_healthcare, 2021, doi:10.3390/healthcare9121619_

Round 1
Reviewer 1 Report
This evaluation of data in the public sector to investigate the current status of the epidemiology of scrub typhus can be beneficial to health care providers and policy makers. However, the lack of including a rickettsiologist on the team lowers the content and appropriateness of statements written. I would highly recommend having a professional in the field reread the manuscript before resubmitting the paper.
Please note suggested changes and comments:
Abstract
Lines 15 and 38: Use italics for Orientia tsutsugamush
Key words
Would replace Rickettsia with Orientia tsutsugamushi
Introduction
Line 46: Scrub typhus is a disease and not an agent, therefore I would suggest that you rewrite this sentence to say that “, Scrub typhus, cause by infection with Orientia species, is the most common rickettsial disease in the world.”
Line 48: capitalize “Gram” when using his name when describing a stain that he discovered. Would indicate that the orientiae are parasitic to their “host cells”.
Line 52: Need to indicate that only O. tsutsugamushi causes scrub typhus in the Tsutsugamushi Triangle, and that other species of Orientia cause scrub typhus in other locations, such as South America, Africa, and the Middle East.
Line 59: please do not indicate that tsutsugamushi is part of the genus Rickettsia. Orientia tsutsugamushi, definitely part of the family Rickettsiaceae and the order Rickettsiales, but has not be considered part of the genus Rickettsia since 1995.
Line 60: Again, scrub typhus is not an agent, so you cannot become infected with scrub typhus.
Line 67: Chigger is a stage of mite, so saying chigger mites means what? Infection occurs following the bite of the chigger stage of an infected mite might be what you are trying to get at.
So when you are talking about mites in general, do not write chigger mites.
Line 70: chigger species makes no sense what so ever.
Line 72: what do you mean the animal host of tsutsugamushi. Are you saying that “tsutsugamushi” is a disease, an agent, or what?
Line 73: No rodents are not the host for Orientia spp. Their host are the mites. However, the mites’ host are the rodents.
Line 75: Again stop using the genus Rickettsia when talking about the genus Orientia.
Line 75: the word for the agent passed from mother to daughter cells in the egg is transovarial transmission and not: “pass” through the “egg heredity”.
Line 76: Stop using the genus name Rickettsia unless it is appropriate.
Line 77: life history of a chigger is a single stage in a mite life cycle.
Line 81: Chigger mites at the nymph and adult stage makes no sense.
Line 82 : what is tsutsugamushi infection?
Line 86: at the end of the sentence indicate that . . . are the main rodent hosts for the infected mites.
Line 90: what do you mean, prevalence of tsutsugamushi?
Line 92 and 94: once you have started using the abbreviation O. tsutsugamushi, it is no long er necessary to write out the genus species name, i.e. Orientia tsutsugamushi.
Line 111: depending on the strain of O. tsutsugamushi subsequent infections can be quite sever and even life threatening if not treated properly.
Author Response
Dear the reviewer, November 4, 2021
We resubmitted the manuscript entitled “Epidemiology and Risk Factors for Notifiable Scrub Typhus in Taiwan during the period 2011-2019” to the Journal after amendments made based on reviewers comments. We have carefully revised our manuscript according to reviewer’s critiques and suggestions. We marked amendments in yellow font in the manuscript for clarity purpose. Our specific responses to reviewer’s comments are as follows.
Reviewer 1
This evaluation of data in the public sector to investigate the current status of the epidemiology of scrub typhus can be beneficial to health care providers and policy makers. However, the lack of including a rickettsiologist on the team lowers the content and appropriateness of statements written. I would highly recommend having a professional in the field reread the manuscript before resubmitting the paper.
Response: Thanks the reviewer comment. We have consulted rickettsiologist from Institute of Pathology and Parasitology, National Defense Medical Center, Taiwan, which having a professional in the field to reread the revised manuscript before the authors resubmitting the paper. The authors have also revised all the reviewer’s recommendations.
Please note suggested changes and comments:
Abstract
- Lines 15 and 38: Use italics for Orientia tsutsugamush
Response: Thanks the reviewer comment. The authors have revised the word “Orientia tsutsugamush” as line 15 and 38.
Key words
- Would replace Rickettsia with Orientia tsutsugamushi
Response: Thanks the reviewer comment. The authors have replaced “Rickettsia with Orientia tsutsugamushi” as line 41.
Introduction
- Line 46: Scrub typhus is a disease and not an agent, therefore I would suggest that you rewrite this sentence to say that “, Scrub typhus, cause by infection with Orientiaspecies, is the most common rickettsial disease in the world.”
Response: Thanks the reviewer comment. The authors have revised the sentence “Scrub typhus, cause by infection with Orientia species, is the most common rickettsial disease in the world.” as line 46 and 47.
- Line 48: capitalize “Gram” when using his name when describing a stain that he discovered.Would indicate that the orientiae are parasitic to their “host cells”.
Response: Thanks the reviewer comment. The authors have revised the sentence “…, a Gram-negative bacterium that is intracellular parasite.” as line 49.
- Line 52: Need to indicate that only tsutsugamushi causes scrub typhus in the Tsutsugamushi Triangle, and that other species of Orientiacause scrub typhus in other locations, such as South America, Africa, and the Middle East.
Response: Thanks the reviewer comment. The authors have added the sentence “Only O. tsutsugamushi causes scrub typhus in the Tsutsugamushi Triangle, and that other species of Orientia cause scrub typhus in other locations, such as South America, Africa, and the Middle East.” as line 55-57.
- Line 59: please do not indicate that tsutsugamushiis part of the genus Orientia tsutsugamushi, definitely part of the family Rickettsiaceae and the order Rickettsiales, but has not be considered part of the genus Rickettsia since 1995.
Response: Thanks the reviewer comment. The authors have deleted sentence “Rickettsia” as line 62.
- Line 60: Again, scrub typhus is not an agent, so you cannot become infected with scrub typhus.
Response: Thanks the reviewer comment. The authors have replaced “scrub typhus” to “O. tsutsugamushi” as line 63.
- Line 67: Chigger is a stage of mite, so saying chigger mites means what?Infection occurs following the bite of the chigger stage of an infected mite might be what you are trying to get at. So when you are talking about mites in general, do not write chigger mites.
Response: Thanks the reviewer comment. The authors have replaced “chigger mites” to “the chigger stage of an infected mites” as line 69.
- Line 70: chigger species makes no sense what so ever.
Response: Thanks the reviewer comment. The authors have deleted the sentence “chigger species” as line 72.
- Line 72: what do you mean the animal host of tsutsugamushi. Are you saying that “tsutsugamushi” is a disease, an agent, or what?
Response: Thanks the reviewer comment. The authors have replaced “tsutsugamushi” to “O. tsutsugamushi infection” as line 73-74.
- Line 73: No rodents are not the host for OrientiaTheir host are the mites. However, the mites’ host are the rodents.
Response: Thanks the reviewer comment. The authors have revised the sentence “The mites host are rodents including rats.” as line 76.
- Line 75: Again stop using the genus Rickettsiawhen talking about the genus Orientia.
Response: Thanks the reviewer comment. The authors have replaced “Rickettsia ” to “Orientia” as line 77-78.
- Line 75: the word for the agent passed from mother to daughter cells in the egg is transovarial transmission and not: “pass” through the “egg heredity”.
Response: Thanks the reviewer comment. The authors have revised the sentence “… through transovarial transmission.” as line 77.
- Line 76: Stop using the genus name Rickettsiaunless it is appropriate.
Response: Thanks the reviewer comment. The authors have replaced “Rickettsia ” to “Orientia” as line 78.
- Line 77: life history of a chigger is a single stage in a mite life cycle.
Response: Thanks the reviewer comment. The authors have revised “The life history of a chigger is a single stage in a mite life cycle” as line 79-80.
- Line 81: Chigger mites at the nymph and adult stage makes no sense.
Response: Thanks the reviewer comment. The authors have deleted “Chigger mites at the nymph and adult stage “ as line 84.
- Line 82 : what is tsutsugamushi infection?
Response: Thanks the reviewer comment. The authors have replaced “tsutsugamushi infection “ to “O. tsutsugamushi infection” as line 84.
- Line 86: at the end of the sentence indicate that . . . are the main rodent hosts for the infected mites.
Response: Thanks the reviewer comment. The authors have revised the sentence “… are the main rodent hosts for the infected mites.” as line 89.
- Line 90: what do you mean, prevalence of tsutsugamushi?
Response: Thanks the reviewer comment. The authors have replaced “prevalence of tsutsugamushi“ to “prevalence of O. tsutsugamushi infections” as line 92.
- Line 92 and 94: once you have started using the abbreviation tsutsugamushi, it is no longer necessary to write out the genus species name, i.e. Orientia tsutsugamushi.
Response: Thanks the reviewer comment. The authors have replaced “Orientia tsutsugamushi“ to “O. tsutsugamushi infections” as line 94 and 96.
Line 111: depending on the strain of O. tsutsugamushi subsequent infections can be quite sever and even life threatening if not treated properly.
Response: Thanks the reviewer comment. The authors have revised the sentence “…, depending on the strain of O. tsutsugamushi subsequent infections can be quite sever and even life threatening if not treated properly.” as line 114-115.
Hopefully, our revised manuscript could fulfill your scientific requirements for publication.
Sincerely yours,
Chia-Peng Yu, Ph.D. (the corresponding author)
School of Public Health,
National Defense Medical Center
No.161 Sec. 6, Minquan E. Rd., Neihu Dist., Taipei 114, Taiwan, Republic of China,
Tel: +886-2-87923311 ext. 16791, Fax: +886-2-87924379,
e-mail: yu6641@gmail.com
Reviewer 2 Report
General comments:
A map of Taiwan and surrounding islands would be helpful in showing the different regions and population densities. Moreover, an incidence map showing the clusters of infection during peaks in 2013 and 2015-2016 would be beneficial.
Specific comments:
Line 15 - Suggest adding “bacterium” in front of Orientia tsutsugamushi.
Line 22 - Change “Risk factors might be” (passive) to “Risk factors were” (active).
Lines 44 to 46 - Reference is required.
Line 48 - Change “bacteria” to bacterium” and “parasitic in the cell” to “intracellular parasite”.
Line 53 - Appears to be all bold font.
Lines 69 to 70 - Distinguish between the phrases “new species” and “newly recorded species”.
Line 73 - Rats carry Orientia tsutsugamushi but it is incorrect to state that they transmit it. This sentence should more clearly sate that transmission occurs when an infected rat is bitten by a chigger mite, which in turn bites an uninfected rat or human.
Lines 79 to 80 - Does the 3-month (89 day) generation time correlate with peak summer and fall months mentioned in the abstract?
Line 86 - Is there an explanation why Rattus tanezumi (house rat) becomes more infected with O. tsutsugamushi than Rattus exclans a more sylvian (grass & woodlands) rodent or Rattus losea (rice-field rat)?
Line 94 - Change “bitten by” to “infected with”.
Line 98 - Describe eschar lesion since it is a major feature of initial OT infection.
Line 143 - “Macular” should be spelled with a lowercase letter “m”.
Line 267 - Should this be “outbreak” instead of “epidemic”?
Line 295 - Change “infectious” to “infected”.
Line 302 - “Main effective factor” would be better stated as “main factor” or “major factor”.
Line 312 - Epidemic may not be the correct term to use. Suggest using “occurrence” or “incidence”.
Line 359 - There appears to be a missing word between “previous” and “showed”?
Lines 380 to 386 - Remove italics.
Lines 446 to 447 - The statement “We purport that humans must consider how to interact with wild animals and find ways to coexist safely with wild animals and disease vectors.” is not logical since rodents (rats) and humans will never achieve this type of relationship.
Author Response
Dear the reviewer, November 4, 2021
We resubmitted the manuscript entitled “Epidemiology and Risk Factors for Notifiable Scrub Typhus in Taiwan during the period 2011-2019” to the Journal after amendments made based on reviewers comments. We have carefully revised our manuscript according to reviewer’s critiques and suggestions. We marked amendments in yellow font in the manuscript for clarity purpose. Our specific responses to reviewer’s comments are as follows.
Reviewer 2
A map of Taiwan and surrounding islands would be helpful in showing the different regions and population densities. Moreover, an incidence map showing the clusters of infection during peaks in 2013 and 2015-2016 would be beneficial.
Response: Thanks the reviewer comment.
Specific comments:
- Line 15 - Suggest adding “bacterium” in front of Orientia tsutsugamushi.
Response: Thanks the reviewer comment. The authors have revised sentense “bacterium Orientia tsutsugamushi ” as line 15.
- Line 22- Change “Risk factors might be” (passive) to “Risk factors were” (active).
Response: Thanks the reviewer comment. The authors have revised “Risk factors might be” to “Risk factors were” as line 23.
- Lines 44 to 46- Reference is required.
Response: Thanks the reviewer comment. The authors have added ”the reference [1]” as line 46.
- Line 48- Change “bacteria” to bacterium” and “parasitic in the cell” to “intracellular parasite”.
Response: Thanks the reviewer comment. Thanks the reviewer comment. The authors have changed “bacteria” to bacterium” and “parasitic in the cell” to “intracellular parasite”. as line 49.
- Line 53- Appears to be all bold font.
Response: Thanks the reviewer comment. The authors have revised “The Tsutsugamushi Triangle”, Which appears to be all bold font as line 53-54.
- Lines 69 to 70- Distinguish between the phrases “new species” and “newly recorded species”.
Response: Thanks the reviewer comment. The author accepted reviewer 1 and reviewer 2 comments. The authors have deleted the sentences “A previous study identified three new species and 23 newly recorded species,…” as line 72.
- Line 73- Rats carry Orientia tsutsugamushi but it is incorrect to state that they transmit it. This sentence should more clearly sate that transmission occurs when an infected rat is bitten by a chigger mite, which in turn bites an uninfected rat or human.
Response: Thanks the reviewer comment. The authors have revised the sentences “Transmission occurs when an infected rat is bitten by a chigger mite, which in turn bites an uninfected rat or human.” as line 75-76.
- Lines 79 to 80- Does the 3-month (89 day) generation time correlate with peak summer and fall months mentioned in the abstract?
Response: Thanks the reviewer comment. The authors have revised the sentences “The 3-month (89 days) generation time may correlate with peak summer and fall months.” as line 843-84.
- Line 86- Is there an explanation why Rattus tanezumi (house rat) becomes more infected with tsutsugamushi than Rattus exclans a more sylvian (grass & woodlands) rodent or Rattus losea (rice-field rat)?
Response: Thanks the reviewer comment. A previous study [1] indicated that among all vertebrate hosts, R. tanezumi had the highest loads and prevalence of chigger infestations, Orientia tsutsugamushi (OT) seropositivity rate in rodents, and OT positivity rate in chiggers. This rodent species was recognized as Rattus rattus before (e.g. [2]); however, R. rattus in Taiwan should instead be R. tanezumi [3]. R. tanezumi commonly occurs near human dwellings in Taiwan, it was trapped only in Lanyu Island. This is due to the fact that we deployed traps in the field but not inside or close to human buildings and Lanyu Island is unusual in that R. tanezumi can be easily found in grassland and forest in Lanyu [4]. High chigger infestation and OT positivity rate in R. tanezumi should reflect a high risk to scrub typhus in Lanyu. Indeed, seropositivity rate for scrub typhus was extremely high (100 %) when children in Lanyu reached 7 years old [5].
Reference:
[1]. Chi-Chien Kuo1, Pei-Lung Lee , Chun-Hsung Chen, Hsi-Chieh Wang. Surveillance of potential hosts and vectors of scrub typhus in Taiwan. Parasit Vectors. 2015 Dec 1;8:611.
[2]. Lin PR, Tsai HP, Weng MH, Lin HC, Chen KC, Kuo MD, et al. Field assessment of Orientia tsutsugamushi infection in small mammals and its association with the occurrence of human scrub typhus in Taiwan. Acta Trop. 2014;131:117–23.
[3]. Musser GG, Carleton MD. Superfamily Muroidea. Pp. 894–1531. In: Reeder DM, Wilson DE, editors. Mammal species of the world: a taxonomic and geographic reference. Baltimore: Johns Hopkins University Press; 2005.
[4]. Shih CD. Forest resource use by Asian house rat (Rattus tanezumi) in Orchid Island. Master thesis. National Dong Hwa University: Institute of Natural Resources Management; 2006. In Chinese, English Abstract.
[5]. Wu BH. An epidemiological study on scrub typhus in kindergarten children in Lanyu. Taiwan Epidemiol Bull. 1993;9:25–30 (In Chinese).
- Line 94- Change “bitten by” to “infected with”.
Response: Thanks the reviewer comment. The authors have replaced ” bitten by” to “infected with” as line 96-97.
- Line 98- Describe eschar lesion since it is a major feature of initial OT infection.
Response: Thanks the reviewer comment. The authors have added the sentence ” An eschar which usually begins as a primary papular lesion later crusts to form a black ulcer with central necrosis is a distinct feature of scrub typhus” as line 101-102.
- Line 143- “Macular” should be spelled with a lowercase letter “m”.
Response: Thanks the reviewer comment. The authors have revised the word “macular” with a lowercase letter “m” as line 147.
- Line 267- Should this be “outbreak” instead of “epidemic”?
Response: Thanks the reviewer comment. The authors replaced ” epidemic“ to “outbreak” as line 276.
- Line 295- Change “infectious” to “infected”.
Response: Thanks the reviewer comment. The authors replaced ”infectious“ to “infected” as line 303.
- Line 302- “Main effective factor” would be better stated as “main factor” or “major factor”.
Response: Thanks the reviewer comment. The authors replaced ” Main effective factor“ to “Main factor ”as line 310.
- Line 312- Epidemic may not be the correct term to use. Suggest using “occurrence” or “incidence”.
Response: Thanks the reviewer comment. The authors replaced the word ”epidemic” to “incidence”, see line 320.
- Line 359- There appears to be a missing word between “previous” and “showed”?
Response: Thanks the reviewer comment. The authors replaced the sentence ”previous showed” to “previous study showed”, see line 367.
- Lines 380 to 386- Remove italics.
Response: Thanks the reviewer comment. The authors removed italics of the manuscript, see line 389-396.
- Lines 446 to 447- The statement “We purport that humans must consider how to interact with wild animals and find ways to coexist safely with wild animals and disease vectors.” is not logical since rodents (rats) and humans will never achieve this type of relationship.
Response: Thanks the reviewer comment. The authors deleted the sentences “We purport that humans must consider how to interact with wild animals and find ways to coexist safely with wild animals and disease vectors.”, see line 456.
Hopefully, our revised manuscript could fulfill your scientific requirements for publication.
Sincerely yours,
Chia-Peng Yu, Ph.D. (the corresponding author)
School of Public Health,
National Defense Medical Center
No.161 Sec. 6, Minquan E. Rd., Neihu Dist., Taipei 114, Taiwan, Republic of China,
Tel: +886-2-87923311 ext. 16791, Fax: +886-2-87924379,
e-mail: yu6641@gmail.com
Round 2
Reviewer 2 Report
None offered, authors sufficiently addressed all noted issues.
Author Response
Dear the reviewer, November 16, 2021
We resubmitted the manuscript entitled “Epidemiology and Risk Factors for Notifiable Scrub Typhus in Taiwan during the period 2010-2019” to the Journal after amendments made based on reviewers comments. We have carefully revised our manuscript according to reviewer’s critiques and suggestions. We marked amendments in yellow font in the manuscript for clarity purpose. Our specific responses to reviewer’s comments are as follows.
Reviewer 2
None offered, authors sufficiently addressed all noted issues.
Response: Thanks the reviewer comment. We also had fine/minor spell correction for English language and style as line 121, 221, 248, 266, 288, and 412.
Hopefully, our revised manuscript could fulfill your scientific requirements for publication.
Sincerely yours,
Chia-Peng Yu, Ph.D. (the corresponding author)
School of Public Health,
National Defense Medical Center
No.161 Sec. 6, Minquan E. Rd., Neihu Dist., Taipei 114, Taiwan, Republic of China,
Tel: +886-2-87923311 ext. 16791, Fax: +886-2-87924379,
e-mail: yu6641@gmail.com
